# Examining Decomposition and Nitrogen Mineralization in Five Common Urban Habitat Types across Southern California to Inform Sustainable Landscaping

George L. Vourlitis [1,*], Emma Lousie van der Veen [2], Sebastian Cangahuala [3], Garrett Jaeger [1], Colin Jensen [1], Cinzia Fissore [4], Eric M. Wood [5], Joel K. Abraham [6], Kevin S. Whittemore [6], Elijah Slaven [7], Dustin VanOverbeke [8], James Blauth [8], Elizabeth Braker [9], Nina Karnovsky [2] and Wallace M. Meyer III [2,*]

[1] Department of Biological Sciences, California State University, San Marcos, CA 92096, USA
[2] Biology Department, Pomona College, Claremont, CA 91711, USA
[3] Claremont High School, Claremont, CA 91711, USA
[4] Department of Biology and Environmental Science, Whittier College, Whittier, CA 90608, USA
[5] Department of Biological Sciences, California State University Los Angeles, Los Angeles, CA 90032, USA
[6] Department of Biological Science, California State University, Fullerton, CA 92831, USA
[7] Sue and Bill Gross School of Nursing, University of California, Irvine, Irvine, CA 92697, USA
[8] Biology Department, University of Redlands, 1200 E Colton Ave., Redlands, CA 92373, USA
[9] Biology Department, Occidental College, 1600 Campus Road, Los Angeles, CA 90041, USA
* Correspondence: georgev@csusm.edu (G.L.V.); wallace_meyer@pomona.edu (W.M.M.III)

**Abstract:** Urban landscaping conversions can alter decomposition processes and soil respiration, making it difficult to forecast regional $CO_2$ emissions. Here we explore rates of initial mass loss and net nitrogen (N) mineralization in natural and four common urban land covers (waterwise, waterwise with mulch, shrub, and lawn) from sites across seven colleges in southern California. We found that rates of decomposition and net N mineralization were faster for high-N leaf substrates, and natural habitats exhibited slower rates of decomposition and mineralization than managed urban landcovers, especially lawns and areas with added mulch. These results were consistent across college campuses, suggesting that our findings are robust and can predict decomposition rates across southern California. While mechanisms driving differences in decomposition rates among habitats in the cool-wet spring were difficult to identify, elevated decomposition in urban habitats highlights that conversion of natural areas to urban landscapes enhances greenhouse gas emissions. While perceived as sustainable, elevated decomposition rates in areas with added mulch mean that while these transformations may reduce water inputs, they increase soil carbon (C) flux. Mimicking natural landscapes by reducing water and nutrient (mulch) inputs and planting drought-tolerant native vegetation with recalcitrant litter can slow decomposition and reduce regional C emissions.

**Keywords:** carbon cycle; C emissions; native plant; nitrogen cycle; urban ecology; Mediterranean; *Qurecus argrifolia*; *Plantanus recemosa*

## 1. Introduction

Covering less than 2% of Earth's surface, Mediterranean climate ecosystems are home to more than 250 million human residents [1]. Because Mediterranean climate ecosystems harbor approximately 20% of the world's vascular plant species, the resulting development and habitat conversions—via the process of urbanization—associated with large and expanding human populations in these regions pose a significant threat to global biodiversity [2–6]. In addition to its impacts on biodiversity, urbanization can also alter key ecosystem processes [7–12]. For example, decomposition and soil respiration represent the second-largest carbon (C) flux to the atmosphere [13]. Therefore, studies that explore how urban landscaping modifications affect litter decomposition processes are needed to improve our

ability to predict and potentially mitigate greenhouse gas emissions in urban areas, particularly those in Mediterranean climate ecosystems heavily impacted by human activities.

Because urbanization influences multiple drivers of decomposition (temperature, humidity, litter quality, and decomposer community), it is difficult to predict how urban land uses alter this critical ecosystem process [9,12,14–18]. For example, urban warming, soil disturbance, fertilization, nitrogen (N) deposition, and alterations to water infiltration and/or water holding capacity all can influence rates of microbial respiration and decomposition [9,12,18,19]. Vegetation composition also plays a key role in litter decomposition due to species differences in litter quality (ca. C:N ratio, lignin content) and species effects on soil fauna and microbial communities [14,16,18]. Finally, urbanization creates novel environments, such as exotic vegetation and interfaces that can alter hydrology, introduce toxins, and alter nutrient inputs, which can further affect rates of microbial growth, activity, and as a result, litter decomposition [12,17]. Thus, to accurately model regional urban C dynamics, we must understand how various landscape modifications affect decomposition processes.

Addressing the impacts of urbanization on decomposition is particularly important in southern California and northern Mexico, which have more than double the human population density of any other Mediterranean region [5,6]. The human population in this region continues to increase and expand into natural areas [20–22]. In addition, this region's human population is disproportionately concentrated in lowland areas, much of which are now urban or otherwise disturbed [5]. Lowland areas in southern California and northern Mexico were once covered by grasslands, riparian forests, oak woodlands in foothill regions, and wetlands, but California sage scrub (hereafter, sage scrub) was and remains the most common native habitat [23,24]. Currently, sage scrub is an endangered shrub-dominated ecosystem type that has been reduced to less than 10% of its original range, with many areas having been converted either to non-native grasslands or urban/suburban areas [5,6,24–26]. Studies in southern California have shown that rates of decomposition and microbial respiration are elevated and soil C-storage is reduced in non-native grasslands relative to areas with native sage scrub, indicating that this habitat modification will lead to increased regional flux of C into the atmosphere [27–30]. However, the semiarid climate of southern California, coupled with the unique landscape management of urban environments (e.g., use of "waterwise" shrubs and mulches to reduce water use and losses), has the potential to alter decomposition rates compared to patterns observed in native vegetation and non-native grasslands. For example, differences in vegetation (turfgrass vs. xeric shrubs) or mulching may alter the temperature or moisture sensitivity of soil microbes and decomposition. Additionally, wetting and drying cycles are more frequent in landscaped vs. natural areas and can fundamentally alter rates of decomposition and $CO_2$ emission [31]. Unfortunately, these impacts have been understudied in semiarid regions, particularly southern California.

Here we quantified early (first few months when labile substrates are present) decomposition rates and mass loss for native plant leaf litter decomposing in natural and urban landcovers on seven college campuses across southern California. At each college campus we incubated standardized litter in natural areas or landscaped areas consisting of shrub beds, lawns, or waterwise plantings that were either mulched or without mulch. College campuses are ideal for examining how urban modifications influence ecosystem processes. They harbor most landcover types found across the urban–suburban matrix and have relatively good records of watering and landscape maintenance schedules and materials. Because urban landcovers receive inputs of water, mulch, or fertilizers, which can enhance rates of litter decomposition [32,33], we hypothesized that: (1) rates of mass loss are higher in urban landcovers than in natural vegetation, and (2) higher-quality litter (higher N concentrations and lower C:N and lignin:N ratios) decompose faster. We predicted that decomposition is elevated in irrigated landcovers because soil microbial activity in semiarid regions is often limited by soil moisture [34–36].

## 2. Materials and Methods

### 2.1. Site Descriptions

We measured rates of leaf litter decomposition and nitrogen (N) mineralization between 28 January and 17 May 2019, for two dominant native trees of the southern California region: coastal live oak (*Quercus agrifolia* Neé, hereafter "oak") and western sycamore (*Platanus racemosa* L., hereafter "sycamore") in natural and urbanized landcovers at seven college campuses in southern California (Figure 1, Table A1). These species were selected because they are typical of native woodlands and riparian forests, respectively, commonly found in sage scrub fragments, and are often used as landscape trees in urban greenspaces of southern California. Colleges were located along a ca. 150 km north–south and a 50 km east–west gradient, with Occidental College the northern- and westernmost college, CSU San Marcos the southernmost college, and the University of Redlands the easternmost college (Table A1). The average annual temperature ranged from 16.6 °C at Pomona College to 19.7 °C at Whittier College, and average annual rainfall was highest (451 mm) at CSU Los Angeles and lowest (332 mm) at CSU San Marcos. According to the UC Davis Soil Web (https://casoilresource.lawr.ucdavis.edu/gmap/; accessed 12 May 2022), soils at each site were Entisols; however, suborders varied between typic xerothents (CSU San Marcos, Whittier College, and Occidental College), typic xerofluents (Pomona College, CSU Los Angeles, and CSU Fullerton), and typic xeropsamments (University of Redlands).

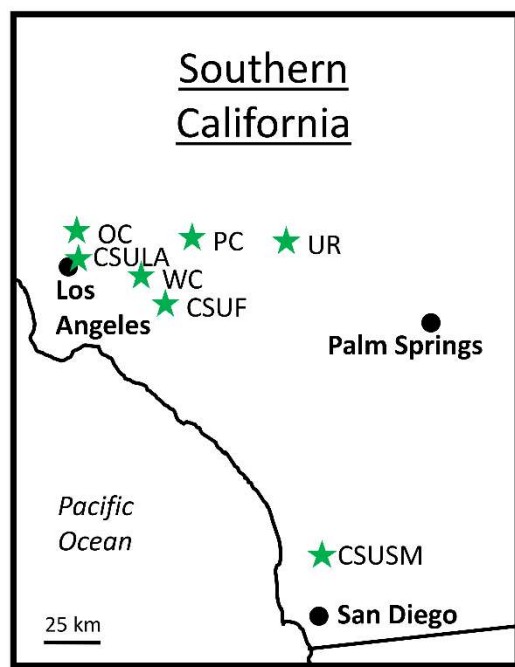

**Figure 1.** Map of the of the seven college campuses in southern California. At each campus, decomposition and nitrogen mineralization was examined in five landcover types. For a description of site characteristics for the seven colleges, see Table A1.

To examine how landcover modifications influence litter decomposition rates in urban/suburban southern California, we placed litter bags (eight of each litter type) in five landcover types at each college that typify southern California urban landscapes: (1) lawns, dominated by non-native grasses and high water inputs; (2) hedge/shrub environments, defined by at least 75% cover of non-native shrubs with water inputs; (3) waterwise gardens (no mulch), which consist of native or non-native plants with reduced water inputs but no mulch; (4) waterwise (with mulch) gardens, with native or non-native plants, reduced water inputs and a mulch (most often mixed woodchips) covered soil; and (5) natural areas, receiving no water subsidies and often dominated by native shrubs or non-native annuals (Figure 2).

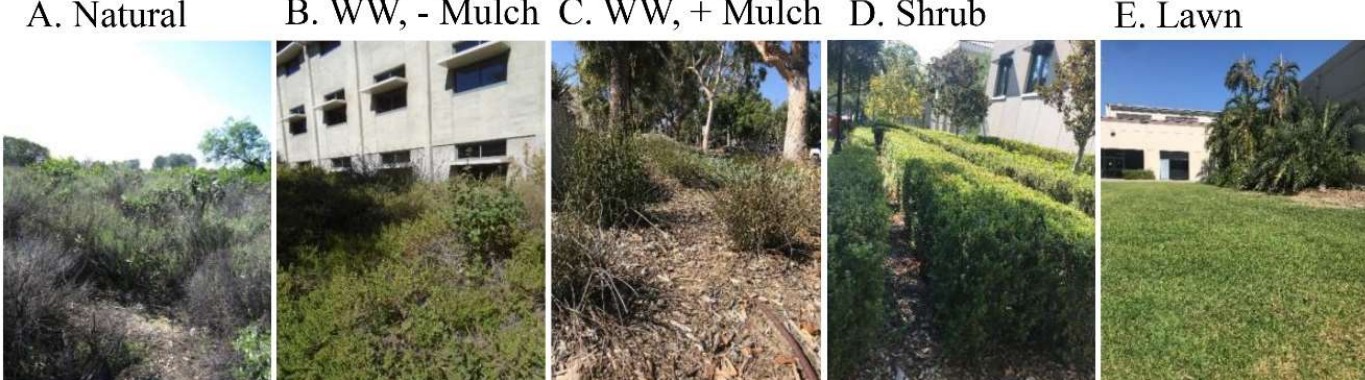

**Figure 2.** Pictures of the five landcover types at Pomona College in Claremont CA: (**A**) natural areas that have no water subsidies, but may include native sage scrub or a combination of native and non-native plants; (**B**) waterwise (WW) areas without added mulch. These habitats varied significantly among campuses, but were similar in that they were landscapes with water-tolerant plants to reduce water subsidies; (**C**) waterwise areas with mulch; (**D**) shrub areas with hedges and other, mostly non-native shrubs, are featured; (**E**) lawns. For a description of site characteristics for each habitat across the seven colleges, see Supplemental Table S1.

Vegetation characteristics within each vegetation type varied by campus (see Supplemental Table S1 for descriptions of habitats within each site). For example, natural areas were not always composed of native species but were areas where no additional management (i.e., irrigation, fertilization, or other soil amendments) occurred. Lawns were generally composed of Kikuyu or Bermuda grass and were irrigated 2–4 times/week. Shrub landscape areas were generally composed of non-native shrubs that were watered 2–3 times per week. Waterwise landscapes varied between succulent gardens (Occidental College, CSU San Marcos, and CSU Los Angeles), native plant gardens (Pomona College and Whittier College), and an orange grove (University of Redlands), but the common feature was limited irrigation. Waterwise landscapes with mulch were similar in vegetation composition and irrigation schedule to waterwise landscapes, but mulch in the form of woodchips or bark was present to reduce water loss.

### 2.2. Litter Decomposition

We collected oak leaves from trees at the Bernard Field Station in Claremont California (34°06′32.96″ N: 117°42′42.47″ W), and sycamore leaves from recently senesced trees on the Pomona College campus. The oak leaves were still attached to the stems at the time of collection and were green in color and not senescent, while the sycamore leaves were attached to the stems at the time of collection but were yellow-brown and senescent (e.g., collected on the ground). While these initial differences in leaf age limit our ability to compare oak and sycamore decomposition kinetics per se, differences between the two substrate types in terms of substrate quality were large (Table 1) and allowed for the analysis of interactions between substrate quality and urban land cover. We air-dried (ca. 30 °C) litter and placed 1.5 to 2.0 g of the air-dried litter in litter bags constructed of 2.0 mm mesh mosquito netting. We oven-dried a small subset of leaves (50 °C) to determine the fraction of water remaining after air drying.

We placed eight litter bags of each litter type in each landcover type (80 litter bags per college; 560 total) in the upper 10 cm soil layer, which was associated with the soil A-horizon in typical southern California soils [27]. Burial of litter bags is often done in decomposition studies where active management or other processes result in the loss or burial of litter bags and provides insights into below-ground decomposition processes that often differ from those on the surface [37,38]. In our experiment, disturbance of litter bags was likely because surface litter in most urban landcovers is often removed by raking, mowing, and/or leaf

blowers during routine landscape maintenance [9]. Furthermore, surface litter inputs in urban settings are often buried by soil, mulches, and other organic inputs [9]. While potential mass loss due to radiation exposure is eliminated [30,39], the upper 10 cm soil layer is within the active zone for root growth, microbial activity, and soil organic matter decomposition [37,40,41], and is a preferred location to quantify the impacts of both soil resource availability and substrate quality on decomposition [42]. We deployed litterbags in the field between 28 January and 27 February 2019 and retrieved them between 1 May and 17 May 2019, resulting in a 72–97 day incubation period depending on the campus. This study was mainly conducted during the cool-moist Mediterranean season. This season was intentionally chosen because: (1) the cool-moist Mediterranean season coincides with the spring semester at our participating institutions with many scheduled ecology-focused classes, which we used to assist in the deployment and recovery of the litter bags, and (2) it enabled us to conservatively examine differences among urban landscapes because soil moisture differences would likely be exacerbated during the hot-dry Mediterranean season (June to October), particularly with regard to comparisons between natural and irrigated urban habitats.

**Table 1.** Mean (±se) initial leaf carbon and nitrogen chemistry for the oak and sycamore leaves used in the decomposition experiment. Shown also are the results of a 2-sample *t*-test (with degrees of freedom) and a bootstrap randomization test, which were used because of unequal sample sizes (n). All values except the leaf C:N ratio are in percentages. NS = not statistically significant ($p > 0.05$).

| Variable | Oak | n | Sycamore | n | Statistic ($p$) |
|---|---|---|---|---|---|
| Leaf N | $1.51 \pm 0.04$ | 11 | $1.15 \pm 0.05$ | 11 | $t_{20} = 6.2$ ($p < 0.001$) |
| Leaf C | $46.3 \pm 0.3$ | 11 | $43.8 \pm 0.3$ | 11 | $t_{20} = 5.5$ ($p < 0.001$) |
| C:N | $30.7 \pm 1.9$ | 11 | $38.7 \pm 1.6$ | 11 | $t_{20} = -4.6$ ($p < 0.001$) |
| Soluble C | $47.3 \pm 3.5$ | 11 | $35.8 \pm 5.6$ | 6 | Bootstrap (NS) |
| Holocellulose C | $13.7 \pm 3.3$ | 11 | $12.7 \pm 6.4$ | 6 | Bootstrap (NS) |
| Lignin C | $39.0 \pm 5.6$ | 11 | $51.3 \pm 10.8$ | 6 | Bootstrap (NS) |
| Lignin:N | $26.3 \pm 4.5$ | 11 | $46.4 \pm 8.5$ | 6 | Bootstrap ($p < 0.05$) |

*2.3. Laboratory Analyses*

We cleaned harvested litterbags by hand to remove debris and dried them at a temperature of 40–50 °C for three days [37]. We carefully removed dried litter from the litterbags and then weighed them on a digital balance to the nearest 0.01 g and ground the litter to a fine powder using a mill (Wiley 40 mesh) and a ball mill (MM200, Retsch, Dusseldorf, Germany). We subjected a subset of each ground sample to loss on ignition at 550 °C in a muffle furnace for 24 h to determine the fraction of sediment that was mixed in the final litter sample and to correct the estimate of mass loss due to sediment contamination [43]. These corrections are necessary for litter decomposition studies where soil contamination is likely [42–45].

We measured C and N concentrations in the litter before and after incubation using dry combustion (ECS 4010, Costech Analytical Technologies, Inc., Valencia, CA, USA, and an Elementar vario Micro cube analyzer, Elementar Inc., Mt Laurel, NJ, USA). We measured the initial fraction of soluble C, holocellulose, and lignin using methods described by Moorhead and Reynolds (1993). We took two adjacent soil samples from the upper 10 cm soil layer (A-horizon) from each landcover type at each institution during litter bag installation (n = 2/landcover type/institution). The first soil sample consisted of approximately 30 ml of soil that was used to analyze total C and N by elemental analysis (Elementar vario MICRO cube analyzer, Elementar, Mt. Laurel, NJ, USA). The second sample consisted of approximately 250 ml of soil that was analyzed for organic matter content, cation exchange capacity (CEC), pH, and texture at the UC Davis Analytical Laboratory (Davis, CA, U.S.A.).

Soil temperature/humidity was measured in the upper 0–10 cm soil layer every 30 min from each landcover type at three of the seven colleges (Occidental College, University of

Redlands, and CSU Fullerton) using Log Tag HAXO-8 temperature/humidity sensors (n = 1 sensor/landcover).

### 2.4. Data Analyses

We calculated the percentage of mass (or nitrogen) remaining as $M_f/M_o$ (or $N_f/N_o$) $\times$ 100, where $M_f$ and $N_f$ are the final litter dry mass (M) or N content at the end of the incubation period and $M_o$ and $N_o$ are the initial dry mass and N content of the litter, respectively. Because litter samples were incubated over different intervals on each campus, we calculated the decomposition or net N mineralization rate constant ($k$ and $k_N$, respectively) assuming an exponential mass or N loss curve as $-k_x = LN(X_t/X_o)/t$ [46], where $k_x$ is the mass loss or N mineralization rate constant, $X_t$ and $X_o$ are the final and initial litter dry or N mass, respectively, and $t$ is the incubation time in days. We used the Olson [46] model because it is well established, and we had no reason to believe that the decomposition kinetics of the leaf substrates used here would behave differently than substrates used in other studies.

We assessed differences in soil properties between landcover types using a randomized block ANOVA with campus as the blocking (random) variable and landcover (L) as a fixed effect. We analyzed differences in mass loss, N mineralization, and the $k$ or $k_N$ due to substrate type (S) and L using a randomized-block design with campus as a random variable and S and L as fixed effects. A Tukey–Kramer post hoc test was used to assess differences between means in the event of a significant ($p < 0.05$) ANOVA. We assessed differences in initial litter C, N, and C fractions using a 2-sample *t*-test (litter C, N, and the C:N ratio) or bootstrapping (soluble C, holocellulose, and lignin) if sample sizes were unequal. For the bootstrap, we calculated mean and ±95% confidence intervals from 1000 randomly obtained samples (with replacement) from each C fraction response variable [47]. Means with confidence intervals that did not overlap were taken as significantly ($p < 0.05$) different. We used Pearson correlation to assess relationships between mass and N loss, and soil environmental variables. Data were analyzed using Number Cruncher Statistical Software (NCSS V12, Kaysville, UT, USA, https://www.ncss.com/software/ncss/, accessed on 16 June 2019).

Multi-institution projects such as ours, especially those in urban areas where markings cannot be left, have the potential to introduce high variability in response variables due to data loss. We quantified the amount of data retrieved and the variability in response variables (mass and N loss) within different landcover types. We retrieved more data from natural vegetation (mean 95%; range 88–100%) and shrubs (mean 93%; range 88–100%), e.g., the sites where flags and other markers were permitted, than from other urban landcovers, with lawns having the lowest data retrieval (mean 54%; range 0–100%; Supplemental Table S2). Most of the errors in data retrieval were due to lost bags. Incidents of low sample retrieval caused an increase in the within-landcover variability in mass and N loss. For example, the coefficient of variation (CV = standard deviation/mean $\times$ 100) for mass loss in lawns was > 2 times higher than the CV in natural areas (Supplemental Table S2), and we found a statistically significant negative correlation between the mean data retrieval and the CV for mass loss (r = $-0.65$; $p = 0.04$; n = 10) but not N loss. Regardless, the CV was low for oak mass loss, ranging from 10% for natural areas to 26% for lawns, and even smaller for sycamore mass loss, ranging from 6% for natural areas to 16% for lawns (Supplemental Table S2). The CV was generally higher for N loss, likely due to the higher variability in N than mass loss, ranging from 18% for oak litter in natural areas to 47% for waterwise areas, and 19% for sycamore litter in shrub areas to 34% for waterwise areas with mulching. These low CVs suggest that vegetation or maintenance practices for each urban landcover were similar across campuses and that estimates of mass and N loss within each landcover were robust.

## 3. Results

### 3.1. Initial Substrate Chemistry

Oak leaves had significantly higher N and C concentrations than sycamore leaves, but the C:N ratio of oak leaves was lower than that of sycamore (Table 1). Initial differences in soluble C, holocellulose, or lignin concentrations between oak and sycamore leaves were negligible; however, sycamore litter had a significantly higher initial lignin:N ratio than oak litter (Table 1).

### 3.2. Landcover Variations in Soil Properties

Soil pH, cation exchange capacity (CEC), organic matter content (SOM), and total N and C differed among landcover types, while soil texture (sand, silt, and clay content) and the soil C:N ratio did not differ (Table 2). Natural habitats had significantly lower pH and CEC than the urban landcovers. While SOM and total N were similar for natural and waterwise areas without mulch land covers, SOM was elevated in the waterwise areas with mulch land cover (Table 2). Total C was lowest for natural, shrub, and waterwise areas without mulch land covers, while lawns and waterwise areas with mulch land covers had the highest total N and C of all landcovers (Table 2).

**Table 2.** Mean (±se; n = 7 campuses) soil physical and chemical properties for each landcover type. Shown also are the results of a randomized-block ANOVA (F-statistic, degrees of freedom, and *p*-value) with campus as the blocking (random variable) and landcover type as a fixed effect. In the case of a significant ANOVA, means with a different lower-case letter are significantly ($p < 0.05$) different from each other according to a Tukey–Kramer post hoc test. CEC = cation exchange capacity; SOM = soil organic matter.

| Habitat Type | pH | CEC (meq/100 g) | SOM (%) | Sand (%) | Silt (%) | Clay (%) | Total N (%) | Total C (%) | C:N |
|---|---|---|---|---|---|---|---|---|---|
| Lawn | 7.2 ± 0.2 [a] | 19.7 ± 2.5 [a] | 5.5 ± 1.3 [ab] | 62 ± 5 | 25 ± 3 | 13 ± 2 | 0.27 ± 0.06 [a] | 2.8 ± 0.7 [a] | 10.3 ± 0.6 |
| Natural | 6.3 ± 0.3 [b] | 10.9 ± 2.7 [b] | 2.6 ± 0.2 [a] | 69 ± 4 | 19 ± 2 | 11 ± 2 | 0.11 ± 0.02 [b] | 1.3 ± 0.2 [b] | 15.3 ± 3.7 |
| Shrub | 7.5 ± 0.2 [a] | 14.7 ± 3.2 [a] | 4.4 ± 1.3 [ab] | 68 ± 3 | 20 ± 2 | 12 ± 1 | 0.17 ± 0.06 [b] | 2.7 ± 0.9 [a] | 16.2 ± 1.8 |
| Waterwise | 7.5 ± 0.1 [a] | 14.7 ± 4.1 [a] | 2.7 ± 0.6 [a] | 64 ± 5 | 22 ± 3 | 13 ± 2 | 0.11 ± 0.03 [b] | 1.4 ± 0.4 [b] | 18.5 ± 5.3 |
| Waterwise + Mulch | 7.4 ± 0.1 [a] | 22.7 ± 4.1 [a] | 5.7 ± 1.2 [b] | 64 ± 3 | 23 ± 2 | 13 ± 1 | 0.27 ± 0.07 [a] | 3.8 ± 0.9 [a] | 14.7 ± 1.0 |
| $F_{4,24}$ | 6.41 | 3.74 | 4.01 [†] | 1.08 | 1.36 | 0.53 | 3.88 [†] | 3.92 [†] | 2.58 [†] |
| *p*-value | 0.001 | 0.02 | 0.01 | NS | NS | NS | 0.02 | 0.01 | NS |

[†] LN-transformed.

### 3.3. Litter Decomposition and N Mineralization

The percentage of the initial litter mass remaining differed among landcover types and between litter types, with oak litter decomposing faster than sycamore litter (Figure 2). Decomposition rates expressed as the mass-loss rate constant, *k*, were slowest in natural vegetation for both substrate types (Figure 3). Pairwise comparisons highlight that mass loss and *k* were similar for natural areas, shrubs, and waterwise landcovers, but waterwise with mulch and lawn landcovers had elevated *k* values in comparison to the natural vegetation landcover (Figure 3). In contrast, we found the fraction of initial N mass remaining and the N mineralization rate constant ($k_N$) did not differ between substrates (Figure 4). However, N mass remaining and $k_N$ differed among landcover types. Natural vegetation had the lowest $k_N$ of all landcover types, but there were no differences between natural, shrub, and waterwise land covers (Figure 4).

Temperature ($F_{4,9}$ = 0.63; $p$ = 0.65) and relative humidity ($F_{4,9}$ = 0.00; $p$ = 0.45) did not differ among landscape types.

The *k* for sycamore leaves was positively correlated with silt and total N content and negatively correlated with sand content (Figure 5). The oak *k* value increased as soil CEC, SOM, total N, and total C content increased and declined as the soil C:N ratio increased. Oak litter $k_N$ was not significantly correlated with any of the soil variables, while the sycamore $k_N$ was significantly correlated with the soil C:N ratio (Figure 5). Variations in soil

temperature and humidity in the upper 0–10 cm soil layer between the different landcover types did not influence the decomposition and mineralization rates in any consistent manner (Figure 5).

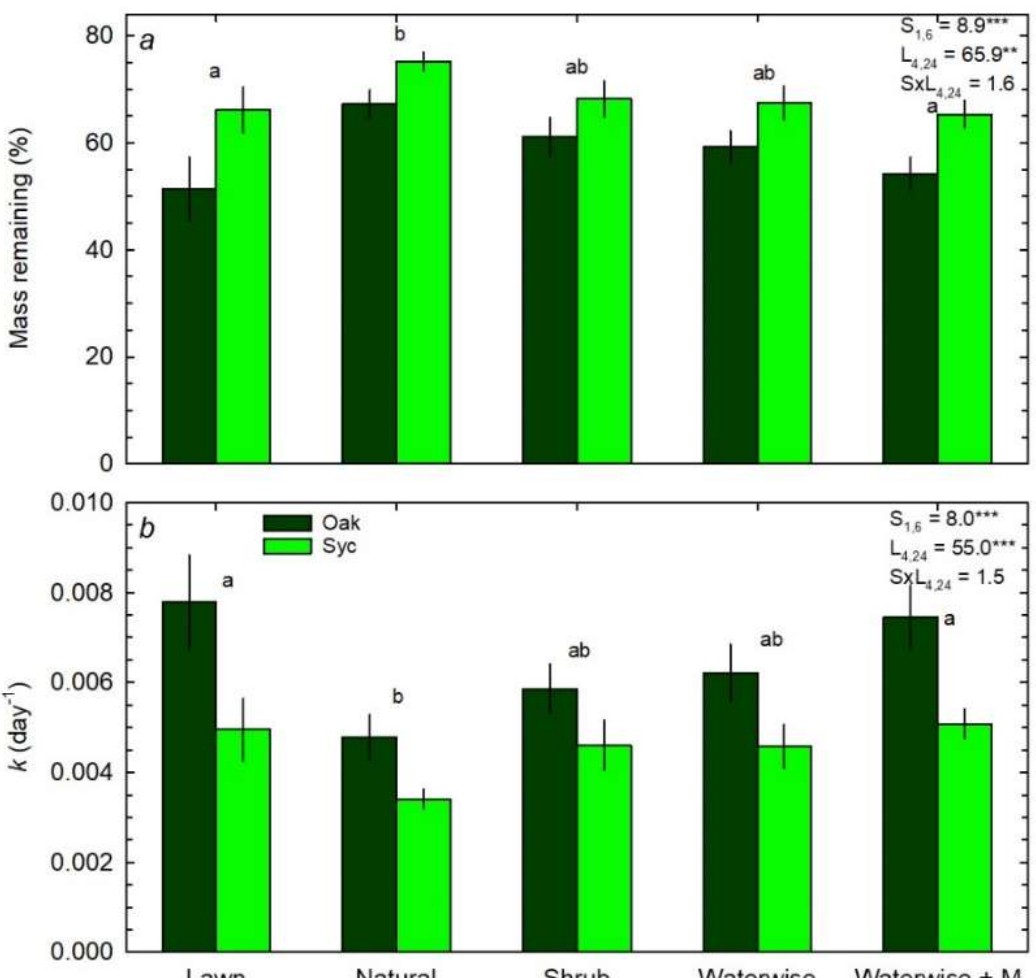

**Figure 3.** Mean (±se; n = 7) percentage of initial mass remaining (**a**) and the mass loss rate constant (*k*) (**b**) from decomposing oak (dark-green bars) and sycamore (light-green bars) leaves. Shown also are the results of a 2-way ANOVA (F-statistic, degrees of freedom, and *p*-value) with leaf substrate (S) and landcover type (L) as fixed effects and the substrate x landcover (S × L) interaction. In the event of a significant ANOVA for site, a Tukey–Kramer post hoc test was run to see which landcovers were significantly different. Means with different letters are significantly different. ** *p* < 0.01; *** *p* < 0.001.

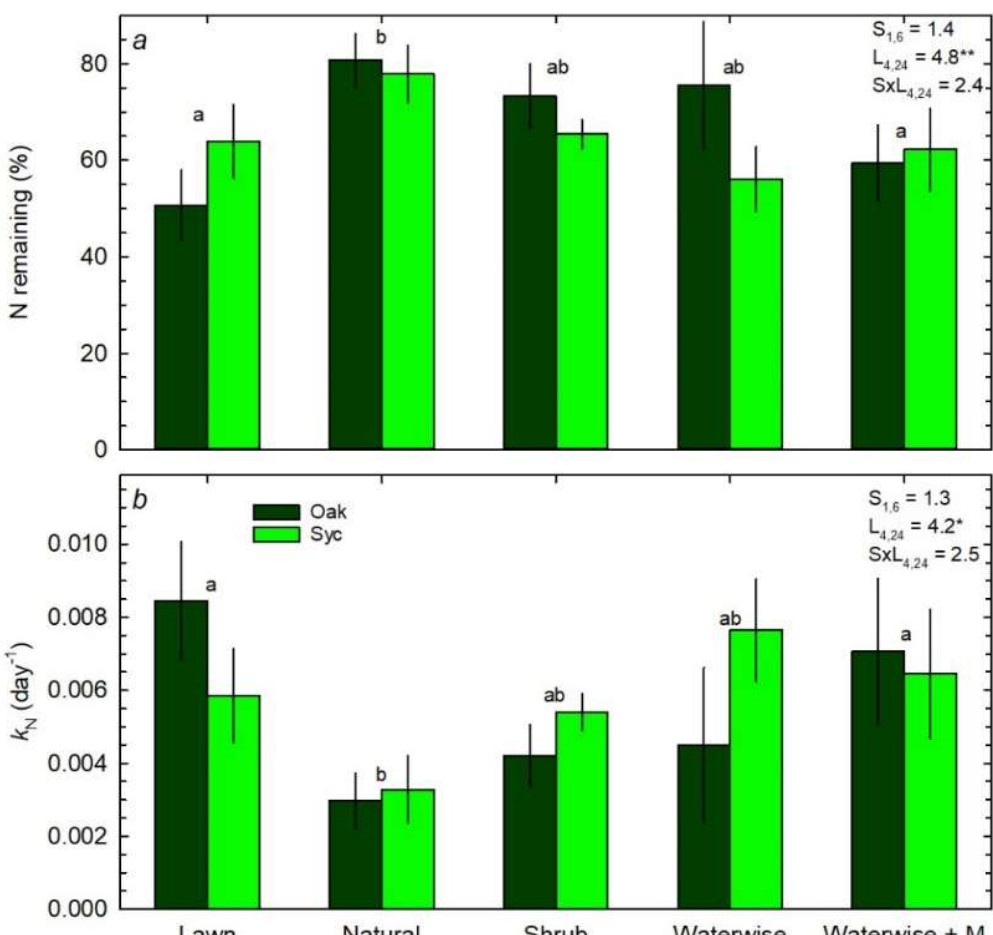

**Figure 4.** Mean (±se; n = 7) percentage of initial litter N remaining (**a**) and the N mineralization rate constant (**b**) from decomposing oak (dark-green bars) and sycamore (light-green bars) leaves. Shown also are the results of a 2-way ANOVA (F-statistic, degrees of freedom, and *p*-value) with species (S) and habitat (H) as fixed effects and the species x habitat interaction. In the event of a significant ANOVA for site, a Tukey–Kramer post hoc test was run to see which sites were significantly different. Means with different letters are significantly different. * $p < 0.05$; ** $p < 0.01$.

| Variable | $k$ (day$^{-1}$) | | $k_N$ (day$^{-1}$) | | |
|---|---|---|---|---|---|
| | Oak | Sycamore | Oak | Sycamore | |
| pH | | | | | 1.00 |
| CEC | ● | | | | 0.80 |
| SOM | ● | | | | 0.60 |
| Sand | | ● | | | 0.40 |
| Silt | | ● | | | 0.20 |
| Clay | | | | | 0.00 |
| Total N | ● | ● | | | -0.20 |
| Total C | ● | | | | -0.40 |
| C:N | ● | | | ● | -0.60 |
| Temperature† | | | | | -0.80 |
| Humidity† | | | | | -1.00 |

**Figure 5.** Pearson-product correlation coefficients for decomposition statistics as a function of soil physical and chemical properties. Cells with a black dot (●) indicate a statistically significant ($p < 0.05$) correlation (n = 35). CEC = cation exchange capacity; SOM = soil organic matter. Data from soil temperature and humidity are from only 3 of the 7 participating institutions (n = 14).

## 4. Discussion

### 4.1. Effects of Landscape Cover on Mass and N Loss

Our data support our hypothesis that rates of litter mass and N loss are higher in urban landcovers than in natural vegetation. The lower rates of decomposition and mineralization in natural areas were expected given that microbial activity, respiration, and decomposition in semiarid southern California soils are often limited by water and/or organic matter inputs [34–36]. However, humidity and temperature did not differ among habitats in the soil A-horizon, suggesting that other factors are driving differences in the decomposition process across these habitats during the cool-wet spring. Humidity and temperature will likely become key drivers of decomposition and elevate differences in decomposition rates between natural and urban areas during the hot-dry Mediterranean season (June through October), when natural areas receive no additional water subsidies.

One possible mechanism explaining the differences between natural and urban habitats during the spring is that native vegetation in semiarid shrublands of southern California (e.g., chaparral and sage scrub) is dominated by woody shrubs that produce litter with high structural C and C:N ratios [48,49], which reduces rates of mass and N loss and causes sage scrub and chaparral soils to be low in SOM, C, and N [50]. In contrast, landscaped areas have "made" soils due to chronic inputs of organic matter, fertilizers, and other soil amendments, irrigation, and the planting of non-native species that can rapidly build up SOM and alter soil physical and chemical properties [9,12,51–55]. These inputs undoubtedly caused SOC, total N, and pH to be higher in nearly all of the landscaped habitats than in the natural habitats observed here. The regular maintenance of these novel habitats reinforces the changes in soil physical and chemical properties over time [12,54], which in turn, alters soil biotic properties [15,18,55] and rates of litter decomposition [9,32,45,56].

We found strikingly similar soil chemical properties and rates of mass loss among the different urban landcovers across the seven campuses. Similar management practices presumably served to homogenize the soil environment within and between urban landcovers. The homogenization of decomposition kinetics or soil properties is likely to occur when human impacts (management practices, disturbance) dominate over natural processes governing soil formation and parent material [19]. Therefore, while spatial variability in urban soils is often predicted to be high, reflecting both natural variability and differences in disturbance intensity and history, watering, fertilizing, and clipping/hedging, we demonstrated relatively consistent patterns in decomposition rates in different urban landscaping types across the region. Thus, we found that these ecological processes and properties can converge within a given landscape type and hypothesized that urban management practices exert strong control over ecosystem processes in urban areas [19]. Because we found that rates of decomposition among habitats were remarkably similar across institutions, data could be used in conjunction with spatial data that quantifies the prevalence of these various habitat types to model greenhouse gas emissions from urban areas in southern California, and explore how modifications to landscaping approaches could reduce or exacerbate C emissions, particularly during the cool-moist season.

### 4.2. Effects of Substrate on Mass and N Loss

We found that the higher-quality oak leaves decomposed faster than sycamore substrate in all landcover types. While this is counter to what is expected given the structure and chemistry of live oak and sycamore leaves [16,48,56–59], at the time of leaf litter harvest, oak litter had a higher tissue N content and a lower C:N and lignin:N ratio than sycamore litter. This is likely because the sycamore litter was senescent at harvest while oak leaves were green and still intact. Mature, healthy green oak leaves have higher N and soluble C than senescent leaves [60], so the foliage quality at the time of harvest likely did not reflect that of senescent oak litter. Our objective was to assess how variations in leaf nutrient concentrations influence decomposition processes, and not how oak and sycamore decomposition rates differ. Our data indicate that rates of mass loss were faster for leaf substrates with higher N content and lower C:N and lignin:N ratios, which is consistent

with previous research. This finding has implications for litter decomposition and C cycling in landscaped areas, since landscaped plants are often supplied with ample N fertilizer. For example, using plants, such as those native to the region, with more recalcitrant litters could slow decomposition processes, possibly enhancing soil carbon storage, and reducing carbon emissions. However, if these plants are regularly fertilized in urban landscapes, their litter will likely be enriched in N, which will stimulate rates of litter decomposition relative to conspecifics in natural vegetation [33,34].

*4.3. Relationship between Environmental Variables and Mass and N Loss*

It is difficult to interpret how microbial activity is affected by environmental characteristics because microbes and soil fauna react to a combination of variables (e.g., temperature, moisture, pH, nutrients, and soil physical properties) that often interact in complex ways [61,62]. We found that rates of oak litter decomposition were positively correlated with SOM, C, and N concentration, while rates of sycamore litter decomposition were positively correlated with silt and total N concentration and negatively affected by sand content. The differences in how litter decomposition responded to environmental variation presumably reflect the differences in initial litter quality [33,34]. First, both litter types exhibited a positive relationship between $k$ and total soil N. Since available N may be positively correlated with total N [63], this implies that mass loss was more rapid in N-rich environments. Increases in N availability have the potential to enhance mass loss of litters rich in cellulose and holocellulose, which is likely during the initial stages of mass loss but inhibit mass loss in heavily lignified substrates [33]. Furthermore, rates of N mineralization may also be higher with high rates of decomposition [43], which would increase available, and presumably total, N. Thus, it is not surprising that initial rates of mass loss for both litter types were positively related to soil N.

We found that soil moisture and temperature had no statistically significant correlation with initial rates of litter mass or N loss. This is surprising given that litter decomposition in semiarid climates may be limited by soil moisture [12,34–36,38]. However, our study was conducted in the spring season, which is typically the coolest and wettest season in southern California, and data indicate that rainfall throughout the study domain was two- to threefold higher during the field study than the long-term average. Thus, soils were equally wet during the study period. However, what is clear is that the landcover variations in $k$ and $k_N$ observed here did not appear to be correlated with variations in soil moisture and temperature, highlighting that during the cool-wet spring other factors are influencing decomposition rates and differences in decomposition between urban and natural areas persist. That is not to say that moisture and temperature are not important factors in litter decomposition in summer and late fall when hot-dry conditions persist. Presumably, landscape variations would increase during the drier part of the year when irrigation of urban landcovers supplements soil moisture, and during dry years in general, when rainfall is below long-term average rainfall for the region. Furthermore, the positive (negative) relationship between soil silt (sand) content and sycamore $k$ may reflect a physical control on soil moisture retention that may influence sycamore litter decomposition [62].

Increases in SOM and SOC also were positively related to oak but not sycamore $k$, which could be due to a variety of processes that influence litter decomposition. For example, higher SOC implies more energy available for soil microbes and fauna that participate in the breakdown of plant litter [54,55]. Caspi et al. [29] found higher bacterial abundance in natural coastal sage scrub (CSS) soils than in soils of non-native grasses across this same region. The higher bacterial abundance was associated with higher SOC content in CSS soils. Microbial functional groups that are associated with C-rich soils are likely copiotrophs, which are rapidly growing microbes that break down more labile types of C [64,65]. This might explain why oak litter, which was higher in N and lower in lignin:N ratio than sycamore litter, decomposed faster in soils with higher SOC.

## 5. Conclusions

Initial rates of litter decomposition and net N mineralization were affected by variations in litter type and urban landscape type. Natural (unmanaged) habitats exhibited slower rates of decomposition and mineralization than managed (irrigated, fertilized landscape), most notably lawns and areas with added mulch. These results were consistent across the seven college campuses in southern California, suggesting that the results found here are robust and part of a general pattern across a semiarid natural-urban gradient.

These results have important implications for C storage in urban greenspaces, particularly those in Mediterranean regions. Native species planting in urban green spaces often leads to improvements in ecosystem services and support of biodiversity [66]. For example, planting native vegetation that has more recalcitrant litter (i.e., woody shrubs, trees) will cause a decline in decomposition and N mineralization, which could enhance soil C and N storage. Using native vegetation will also help minimize the need for water and fertilizer inputs and could also have the same effect on C and N storage, while also simultaneously supporting native animal diversity [67–71]. Thus, planning urban greenspaces that consist of native woody vegetation that needs minimal inputs will likely enhance soil C storage. Interestingly, water input alone may not be a good predictor of decomposition rates, as areas with mulch receive lower water subsidies but have elevated decomposition rates consistent with those found in lawn habitats. This is likely due to the addition of organic C inputs that can enhance decomposer abundance and activity. Consequently, mulched habitats may not be 'sustainable' alternatives to lawns. In southern California, and likely in other Mediterranean regions, landscaping that minimizes C (mulch), water subsidies, and nutrient inputs, and utilizes plants that produce recalcitrant litter can slow decomposition, increase C and N storage, reduce greenhouse emissions, and enhance urban soils as a long-term C and N storage reservoir.

**Supplementary Materials:** The following supporting information can be downloaded at: https://www.mdpi.com/article/10.3390/urbansci6030061/s1, Table S1: Vegetation and management characteristics of the different habitats at each college/university campus; Table S2: coefficient of variation (CV = standard deviation/mean * 100) for the mass and N remaining within each habitat type for oak and sycamore litter.

**Author Contributions:** Conceptualization, G.L.V., C.F., E.M.W., J.K.A., D.V., J.B., E.B., N.K. and W.M.M.III; methodology, G.L.V., C.F., E.M.W., J.K.A., D.V., J.B., E.B., N.K. and W.M.M.III; software, G.L.V. and W.M.M.III; validation, G.L.V., E.L.v.d.V., S.C. and W.M.M.III; formal analysis, G.L.V.; investigation, E.L.v.d.V., S.C., G.J., C.J., K.S.W. and E.S.; resources, G.L.V., N.K. and W.M.M.III; data curation, G.L.V. and W.M.M.III; writing—original draft preparation, G.L.V. and W.M.M.III; writing—review and editing, all authors; visualization, G.L.V. and W.M.M.III; supervision, G.L.V., C.F., E.M.W., J.K.A., D.V., J.B., E.B., N.K. and W.M.M.III; project administration, G.L.V., N.K. and W.M.M.III; funding acquisition, G.L.V., N.K. and W.M.M.III. All authors have read and agreed to the published version of the manuscript.

**Funding:** Funding for this research was provided by Pomona College through a presidential grant, the National Science Foundation (NSF-DBI 2018545 to Meyer and Karnovsky), the Pomona College Biology Department, and the United States Department of Agriculture-National Institute of Food and Agriculture-AFRI (2018-67032-27701 to George Vourlitis).

**Institutional Review Board Statement:** Not applicable.

**Informed Consent Statement:** Not applicable.

**Data Availability Statement:** All raw data and associated metadata are available on the KNB network (https://doi.org/10.5063/F18C9TP1).

**Acknowledgments:** We thank Hans Mickelson (CSUF Landscape Services), Greg Pongetti (Fullerton Arboretum), George Martinez (CSUSM Facilities Management), Isidro Alvarez (CSUSM Facilities Management), and Ronald Nemo (Pomona College Manager of Grounds and Landscaping), and all college/university landscaping crews that helped with landscape maintenance information. We also thank undergraduates who assisted our research, including Chris Clark (Pomona College) and Manny Herrera (Whittier College), the RESCUE-Net community, and the Robert J. Bernard Field Station for access to a research site.

**Conflicts of Interest:** The authors declare no conflict of interest. The funders had no role in the design of the study; in the collection, analyses, or interpretation of data; in the writing of the manuscript; or in the decision to publish the results.

## Appendix A. Location and Abiotic Aspects of the Seven College/University Campuses

**Table A1.** List of participating Colleges with site coordinates and long-term (+30 year) average annual temperature and precipitation for stations closest to the study sites. Climate data are from the Western Regional Climate Center (https://wrcc.dri.edu/summary/Climsmsca.html) accessed on 29 January 2021.

| Site (College) | Latitude (N) | Longitude (W) | Temperature (°C) | Precipitation (mm) |
|---|---|---|---|---|
| Pomona College | 34°05′52″ | 117°42′43″ | 16.6 | 430.5 |
| Whittier College | 33°58′40″ | 118°01′45″ | 19.7 | 375.4 |
| California State University at Fullerton | 33°52′57″ | 117°53′06″ | 17.4 | 365.8 |
| Occidental College | 34°07′38″ | 118°12′36″ | 18.3 | 436.9 |
| University of Redlands | 34°03′46″ | 117°09′48″ | 17.6 | 344.4 |
| California State University at San Marcos | 33°07′46″ | 117°09′35″ | 17.2 | 332.5 |
| California State University at Los Angeles | 34°04′01″ | 118°10′06″ | 18.3 | 451.4 |

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
