# Peer review of "Examining Decomposition and Nitrogen Mineralization in Five Common Urban Habitat Types across Southern California to Inform Sustainable Landscaping"

_urbansci, doi:10.3390/urbansci6030061_

Round 1
Reviewer 1 Report
Line 136: any information on what types of mulch were used? Different types may be very different in terms of C and N content. I see in line 148 that you indicate it was wood chips or bark. Not 100% necessary, but if you have the information, it would be good to say what type of bark (pine, cypress, etc.).
Line 466: you state that mulched landscapes may not affect C and N dynamics. Is this correct? Am I misunderstanding? It seems that the addition of C from mulch does affect C and N dynamics and that this is a key takeaway of your findings.
Author Response
Line 136: any information on what types of mulch were used? Different types may be very different in terms of C and N content. I see in line 148 that you indicate it was wood chips or bark. Not 100% necessary, but if you have the information, it would be good to say what type of bark (pine, cypress, etc.).
The reviewer is correct. The type of mulch does matter. I added what I observed, though we did not collect that data, specifically. However, I know the wood chips used at Pomona College are from mixed tree species collected during landscaping activities and from building materials without nails. In general, the wood chips across sites seemed similar. We did not have any sites with bark nuggets, these are rare, though present, in southern California landscapes. Also, sites did not have any green waste or compost that also can be purchased as a mulch.
Line 466: you state that mulched landscapes may not affect C and N dynamics. Is this correct? Am I misunderstanding? It seems that the addition of C from mulch does affect C and N dynamics and that this is a key takeaway of your findings.
We have edited this section to hopefully eliminate any confusion on what we were trying to indicate. In this instance, we were comparing it to lawns, though I understand that the way it was written may have been confusing.
Reviewer 2 Report
The article “Landscaping matters: Examining decomposition and nitrogen mineralization processes in five common urban habitat types across Southern California to inform sustainable landscaping” addresses the decomposition into various land cover typologies to improve the management of green spaces. It is a well-written and structured article, where the authors present new relevant data. However, I suggest a set of small changes to the document.
The title of the article seems too long. “Landscaping matters” seems to me to be too general, unattractive and does not add information to the type of article.
The Introduction is well developed and presents several relevant works in the area of vegetation and decomposition of organic materials. The objectives are clearly defined, but not very ambitious.
I ask the authors to introduce the classifier in each scientific name of the species. This must be presented at least the first time the name appears in the document.
Line 182-183 - The use of scientists at the beginning of their careers in the implantation and recovery of garbage bags seems to me that it is not very relevant information to present.
The methodology seems adequate to the questions raised by the authors in the Introduction.
The Results are well presented and do not raise relevant doubts.
The Discussion improves the understanding of the results and raises some questions that could be worked on further in the future.
Line 450-452 – This sentence is better suited to the Discussion topic.
The Conclusion is adequate and summarizes the main ideas obtained in the results obtained. However, I think it will be helpful for authors to read a review article on the use of exotic vs native plants: https://doi.org/10.3390/land11081201
Author Response
The article “Landscaping matters: Examining decomposition and nitrogen mineralization processes in five common urban habitat types across Southern California to inform sustainable landscaping” addresses the decomposition into various land cover typologies to improve the management of green spaces. It is a well-written and structured article, where the authors present new relevant data. However, I suggest a set of small changes to the document.
The title of the article seems too long. “Landscaping matters” seems to me to be too general, unattractive and does not add information to the type of article.
Okay, we deleted “Landscaping matters”
The Introduction is well developed and presents several relevant works in the area of vegetation and decomposition of organic materials. The objectives are clearly defined, but not very ambitious.
We understand your perspective, though we cannot add edits to rectify the situation.
I ask the authors to introduce the classifier in each scientific name of the species. This must be presented at least the first time the name appears in the document.
By classifier of scientific names, we interpreted this to mean that the reviewer wanted the authority for the name, but perhaps we misunderstood, but we added that info for each species presented.
Line 182-183 - The use of scientists at the beginning of their careers in the implantation and recovery of garbage bags seems to me that it is not very relevant information to present.
We edited to remove mention of early-career scientists and just linked to using the class to help collect data.
The methodology seems adequate to the questions raised by the authors in the Introduction.
Thank you.
The Results are well presented and do not raise relevant doubts.
Thank you.
The Discussion improves the understanding of the results and raises some questions that could be worked on further in the future.
Line 450-452 – This sentence is better suited to the Discussion topic.
We were a little confused by this request as this topic was discussed in the discussion and re-highlighted in the conclusion. Consequently, we did not make any edits.
The Conclusion is adequate and summarizes the main ideas obtained in the results obtained. However, I think it will be helpful for authors to read a review article on the use of exotic vs native plants: https://doi.org/10.3390/land11081201
Thank you for the recommendation. I think this article supports our conclusions. We added a sentence and cited this manuscript to direct readers interested in this discussion.
Reviewer 3 Report
Dear authors,
It can be seen that the paper is well prepared with a lot of effort. The hypotheses are clearly proved and some good results are carried out. Every section seems OK. Below are a few suggestions for you to consider and revise.
For abbreviation, nitrogen, carbon, or others, please use its full name when it emerges first in the text, then use C and N.
In 2.1 Site descriptions, can you also add a map to indicate the locations of seven campus? And put it before figure 1.
In 2. Materials and Methods, should first generally describe your whole working process, better to add a picture to show the workflow.
Thank you
Author Response
It can be seen that the paper is well prepared with a lot of effort. The hypotheses are clearly proved and some good results are carried out. Every section seems OK. Below are a few suggestions for you to consider and revise.
Thank you.
For abbreviation, nitrogen, carbon, or others, please use its full name when it emerges first in the text, then use C and N.
We added nitrogen and carbon before first use of abbreviations.
In 2.1 Site descriptions, can you also add a map to indicate the locations of seven campus? And put it before figure 1.
We added a map based on your request. However, I am open to removing the map if the editor thinks it is not needed. However, I request to reformat the manuscript before publication if removal of the map is requested.
In 2. Materials and Methods, should first generally describe your whole working process, better to add a picture to show the workflow.
We disagree with this request. Our methods are commonly used in the litter decomposition literature.